# The safety and efficacy of remimazolam tosylate combined with propofol in upper gastrointestinal endoscopy: A multicenter, randomized clinical trial

**Ai Wei[1,2], Shijin Ma[1,2◉], Yuzhe Dou[1,2◉], Xiaojun Wang[3◉], Jianxiong Wu[4◉], Shuzhi Zhou[5◉], Yanfang Deng[6◉], Xinquan Liu[7◉], Dongming Li[8◉], Mengchang Yang[1,2] \***

1 Department of Anesthesiology, Sichuan Provincial People's Hospital, School of Medicine, University of Electronic Science and Technology of China, Chengdu, China, 2 School of Medicine, University of Electronic Science and Technology of China, Chengdu, China, 3 Department of Anesthesiology, Yibin First People's Hospital, Yibin, China, 4 Department of Anesthesiology, Chinese Traditional Medicine Hospital of Leshan, Leshan, China, 5 Department of Anesthesiology, Ya'an People's Hospital, Ya'an, China, 6 Department of Anesthesiology, the first People's Hospital of Liangshan Yi Autonomous Prefecture, Liangshan, China, 7 Department of Anesthesiology, Ziyang People's Hospital, Ziyang, China, 8 Department of Anesthesiology, Bazhong Central Hospital, Bazhong, China

◉ These authors contributed equally to this work.

\* ymc681@126.com

## Abstract

### Introduction

Hypotension is the most common adverse event under propofol-mediated sedation and is possible to cause varying degrees of damage to patients. Whereas remimazolam has a poorer sedative effect than propofol.

### Aim

The aim of this study was to explore the advantages of the combination of remimazolam tosylate and propofol.

### Methods

304 patients were divided into the remimazolam tosylate group (RT group), the propofol group (P group), and the remimazolam tosylate plus propofol group(R+T group). The primary outcome was the incidence of hypotension. Secondary outcomes included the results of sedation and recovery. The safety results mainly include the incidence of Hypotension, adverse respiratory events, postoperative nausea and vomiting, hiccup, cough, body movement and bradycardia.

### Results

The incidence of hypotension was 56.7% in the P group, 12.6% in the RT group, and 31.3% in the R+P group, three groups of pairwise comparisons showed statistical differences, with *P*< 0.001. The incidence of body movement was significantly higher in the RT group

**Data Availability Statement:** All relevant data are within the paper and its Supporting Information files.

**Funding:** The author(s) received no specific funding for this work.

**Competing interests:** The authors have declared that no competing interests exist.

(26.1%) than in the P group (10.3%) and the R+P group (12.5%), $P = 0.004$. The endoscopist satisfaction was higher in the P (3.87±0.44) and R+P (3.95±0.22)groups than in the RT (3.53±0.84) group. The incidence of adverse events, in descending order, was P group, RT group, and R+P group (93.8%vs.61.3%vs.42.7%).

## Conclusion

Co-administration had fewer adverse events than propofol monotherapy, also had a better sedative effect and higher endoscopist satisfaction than remimazolam monotherapy.

## Trial registration

**Clinical trial registration number:** NCT05429086.

## Introduction

Gastric cancer is the fifth most common cancer and the third most common cause of cancer death worldwide, and gastrointestinal endoscopy screening can help identify patients at high risk of developing gastric cancer, thereby reducing the incidence of gastric cancer [1, 2]. However, as an invasive examination, gastrointestinal endoscopy may produce anxiety or discomfort to the patient, and sedation is commonly utilized to reduce disagreeable memory [3, 4]. The sedation protocol recommended by the American Society for Gastrointestinal Endoscopy (ASGE) is benzodiazepines combined with opioids or propofol-mediated sedation [3], whereas previous studies have shown that the sedation of benzodiazepine is less effective and less satisfying for patients than propofol [5], and propofol potentially increases the incidence of hypotension compared to benzodiazepines, although it may improve patient satisfaction [6, 7]. Hypotension is the most common adverse event in sedation with propofol, with an incidence of 31%~35% [8, 9].Recently, perioperative hypotension has been identified as a major factor in some adverse outcomes and should be considered a serious public health issue. Gastrointestinal endoscopy usually requires a longer period of fasting and bowel preparation and is more prone to hypotension. In gastrointestinal endoscopy with propofol sedation, 28% of patients had hypotension lasting longer than 5 minutes and 23% longer than 10 minutes. Few studies have been reported on adverse outcomes due to hypotension of gastrointestinal, but perioperative hypotension lasting longer than 5 minutes may be significantly associated with myocardial and renal injury. Consequently, hypotension associated with gastrointestinal sedation should be emphasized [7, 10, 11].

Remimazolam tosylate (HR7056) is a novel benzodiazepine acting on $GABA_A$ receptors similar to midazolam, however, the utilization of midazolam is limited in outpatient sedation attributed to its metabolites having long-term activity and tend to accumulate, resulting in prolonged recovery time [12]. In contrast, remimazolam tosylate has a rapid onset and is metabolized by unspecific esterases to inactive hydrolysis products [13], leading to a more rapid recovery than midazolam, and is not metabolized by the liver or kidneys, with little inhabitation on circulatory and high safety. Notably, the sedative effect can be expeditiously reversed by flumazenil, which gives remimazolam a unique advantage in outpatient sedation [14, 15].

Several studies have shown that remimazolam is noninferiority to propofol in sedative efficacy and quality of recovery, while significantly reducing the incidence of hypotension and injection pain [9, 16, 17]. Even so, a systematic review and meta-analysis including 1996

patients reported that remimazolam was less effective than propofol in sedation [18], we also found a higher incidence of body movements in patients sedated with remimazolam in a pre-test, suggesting that remimazolam may have poorer sedative than propofol, but has the advantage of having a lower incidence of adverse events and significantly declining the incidence of hypotension, consequently, we considered to combine remimazolam and propofol, also hypothesized that co-administration could decrease the incidence of hypotension, and have a better sedative effect than remimazolam monotherapy.

## Materials and methods

### Study design

This multicenter, randomized, double-blind, positive-controlled trial was initiated by the investigator and approved by the Ethics Committee of Sichuan Provincial People's Hospital (approval number: 420 of 2021, approval date: September 3, 2021). The authors confirm that all ongoing and related trials for this drug/intervention are registered. This trial was registered in the Clinical Trials Center (registration number: NCT05429086). The study protocol conforms to the ethical guidelines of the 1975 Declaration of Helsinki. Patients were recruited from July 01, 2022, to September 30, 2022, the last patient's follow-up was also completed on September 30, 2022. Before beginning enrollment, all investigators received structured education according to the protocol, and quality control officers were assigned to the sub-centers to ensure consistency. Patients provided written informed consent before participation. All study endpoints were collected and analyzed blindly.

### Trial sites and patient population

The trial was conducted in 8 endoscopic centers across Sichuan province. Eligible patients who underwent gastroscopy were evaluated according to the inclusion criteria: 1) aged 18 to 80 years; 2) American Society of Anesthesiologists (ASA) status I or II; 3) Body Mass Index (BMI) 18 to 30 kg/m$^2$. The main exclusion criteria were: 1) preoperative blood pressure higher than 180/110 mmHg or less than 90/60 mmHg; 2) suspected difficult airway and previous history of abnormal anesthesia recovery; 3) severe diseases such as heart, brain, lung, liver, kidney and other organs. Withdrawal criteria: we withdrew patients whose examination time exceeded 15 minutes to reduce the influence of the length of surgery and the dosage of drugs used on the study results.

### Randomization and blinding

After eligibility screening, patients were randomly assigned in a 1:1:1 ratio by centralized randomized grouping, the dosing investigator logged into the randomization system (block of size of 6), and obtained the randomization number, the randomization number generated by SPSS 25.0 software was used as the blind codes to number the study drugs and imported into the centralized randomization grouping system of Sichuan Provincial People's Hospital, drugs were distributed according to the randomization number.

Considering the large difference in the appearance of the trial drugs, remimazolam tosylate (powder) and propofol (emulsion), independent dosing investigators and evaluation investigators were established for this study. The entire trial was blinded to the subjects, but also the evaluation investigator, endoscopist and statistical analysts. The dosing investigator was only involved in the randomization, dosing, and administration process, while the evaluation of the efficacy and safety of the patients and the unscheduled visits were done by the evaluation investigator.

## Interventions

All patients were routinely monitored for ECG, oxygen saturation, and blood pressure after entering the endoscopy room (left lateral position, sphygmomanometer cuff tied to the right hand), given nasal catheter oxygen (4 L/minutes), fentanyl (0.5 µg/kg, Yichang Renfu, China) was first administered intravenously 4 minutes in advance, and infused within 1 minute, followed by the administration of the test drug (test drug administration time was 1 minute): remimazolam tosylate (RT group, 0.2 mg/kg, Jiangsu Hengrui) or propofol (P group, 2 mg/kg, AstraZeneca, UK) or remimazolam tosylate combined with propofol group (R+P group, remimazolam tosylate 0.1 mg/kg + propofol 0.5 mg/kg) for sedation induction, and the examination began after the patients achieved sufficient sedation (MOAA/S score ≤1). During the procedure, the evaluation investigator instructed the dosing investigator to administer the appropriate sedative medication for sedation maintenance based on the MOAA/S score. The examination was initiated when patients achieved adequate sedation; if patients had a MOAA/S score >1 or a MOAA/S score ≤1 but failed to attempt the examination, additional administration of the appropriate sedative drug was allowed 1 minute after the end of the initial dose (RT group: remimazolam tosylate 2.5 mg/dose; P group: propofol 0.5 mg/kg; R+P group: propofol 0.5mg/kg).

Atropine 0.5mg was administered intravenously if the heart rate was less than 45 beats per minute during the examination. When mean arterial pressure (MAP) is less than 70% preoperatively (i.e. hypotension requiring treatment), intravenous rapidly administered crystalloid fluid 200 ml, if blood pressure is still less than 70% preoperatively after volume replenishment, ephedrine 5–10 mg/time can be injected and recorded. The patient is on continuous oxygen during the examination, if pulse oximetry (SpO$_2$) is lower than 95% and time exceeds 30 seconds, jaw thrust was used to improve ventilation; if SpO$_2$ is below 85%, assist ventilation via anesthesia machine or simple respirator, record the change process of oxygen saturation.

## Data collection

From the start of the administration, time to loss of consciousness (LOC), examination start time, additional drug time, examination start time (i.e., the timer started at endoscope removal for 10 minutes and 20 minutes), recovery time (wake-up command at medium volume + tap on the shoulder), the dose of drugs used during the examination and the reason for additional drugs were recorded. Blood pressure, heart rate, respiration, and oxygen monitoring: 3 measurements were taken in the left lateral position within 30 minutes before examination, and the average value was taken as the baseline value. Blood pressure, heart rate, respiration, and oxygen saturation were measured after administration of the test drugs, and then set to be measured and recorded every 2 minutes until the end of the procedure, while the MOAA/S score was evaluated every 3 minutes, and the patient's blood pressure was measured and recorded at the end of the procedure (when the endoscope was withdrawn), recovery time, and every 10 minutes after the procedure until the subject met the criteria for discharge with an Aldrete score of 9 or more reached. Record if first aid treatment is available. Assessment of the quality of recovery included the Steward score, orientation score, head-up assessment, modified Bromage score, and ataxia scale. The evaluation investigators assessed the steward score, orientation, head-up assessment, and modified Bromage score sequentially 30 minutes before the examination, 10 minutes after the end of the examination, and 20 minutes after the examination.

For the collection of adverse events, we defined the adverse events that occurred in this study in a uniform manner. Hypotension was defined as MAP <65 mmHg or a decrease of more than 20% in patients after drug administration, and hypotension requiring treatment

was defined as a decrease of more than 30% in blood pressure; body movement was defined as body movement visible to the naked eye of the investigator; postoperative nausea and vomiting was defined as the direct occurrence of vomiting in patients or verbal reports of nausea by patients; Choking was defined as the patient's choking response during endoscopic insertion; Hiccup was defined as the manifestation of diaphragmatic spasm in patients after medication administration; respiratory adverse events included: oxygen saturation <95%, respiratory count <12 breaths/min, apnea, and upper airway obstruction (defined as abnormal respiratory movements of the patient's thorax directly observed by the investigator), bradycardia was defined as a heart rate <50 beats/min on ECG monitoring, and all adverse events were defined by the investigator.

### Outcomes

The primary outcome was the incidence of hypotension, defined as a mean arterial pressure (MAP) less than 65 mmHg [19, 20], or a decrease in MAP of more than 20% [20].

Secondary outcomes included mainly the incidence of hypotension requiring treatment (defined as a decrease in MAP of more than 30%), duration of hypotension, the success rate of sedation, time to LOC, time to complete recovery (defined as the time between the last dose and the end of examination when the first MOAA/S = 5 at 3 consecutive MOAA/S scores = 5) and quality of recovery, endoscopist satisfaction, anesthesiologist satisfaction, patient satisfaction, and incidence of adverse events.

### Sample size calculation and statistical analysis

The sample size was calculated by PASS 15.0 software. According to our pre-experiment, the incidence of hypotension was found to be 18% in the RT group, 40% in the P group, and 25% in the R+P group, a sample size of 229 achieves 80% power to detect an effect size of 0.2051 using a 2 degrees of freedom Chi-Square Test with a significance level (alpha) of 0.05, and considering the loss rate of 20%, the final sample size should not be less than 287.

Results were analyzed by SPSS 25.0, with quantitative data expressed as mean ± standard deviation and qualitative data expressed as number and frequency. For population baseline analysis, we used one-way analysis of variance (ANOVA) for quantitative data with normal distribution and homogeneous variance, and $\chi^2$ test was used for qualitative data. The primary outcome was tested with the $\chi^2$ test for pairwise comparisons; one-way ANOVA, nonparametric tests (Kruskal-Wallis test), and the $\chi^2$ test were used in the analysis of the secondary outcomes. In one-way ANOVA, if the difference was significant, the Student-Newman-Keuls q test was further used for pairwise comparison of each group.

A two-sided test was chosen for all analyses and was statistically significant at $P< 0.05$. In $\chi^2$ test, multiple comparisons of the enumeration data (P group to R group, P group to R+P group, R group to R+P group) were performed, and the α lever was set at 0.017, following Bonferroni adjustment.

## Results

### Patients

Of 401 patients screened, 29 were excluded because their BMI did not meet the inclusion criteria, 16 failed screening due to systolic blood pressure > 180 mmHg or diastolic blood pressure > 100 mmHg, 5 failed due to heart rate < 50 beats/minutes, 24 withdrew from the study because the examination time exceeded 15 minutes, 9 withdrew due to withdrawal of informed consent, and 14 were lost to follow-up, 304 patients were finally included, including

111 in the RT group, 97 in the P group, and 96 in the R+P group (Fig 1). The baseline characteristics of the included patients and the duration of examination in the three groups did not reach statistical differences. (Table 1).

## Primary outcome

Among patients with a 20% decrease in MAP or MAP < 65 mmHg after administration, the highest incidence was in the P group (n = 55, 56.7%), followed by the R+P group (n = 30,

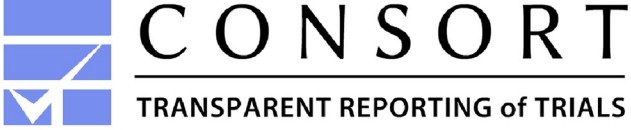

### CONSORT 2010 Flow Diagram

**Enrollment**

Assessed for eligibility (n=401)

Excluded (n=49)
- Not meeting inclusion criteria (n=29  )
- Declined to participate (n=9 )
- Other reasons (n=11  )

**Allocation**

Randomized (n=352  )

RT group (n=117)

P group (n=117)

R+P group (n=117)

**Follow-Up**

Lost to follow-up (n= 1 ),
Withdraw inform content(n=2)
Procedure longer than 15mins(n=3)

Lost to follow-up (n= 16),
Withdraw inform content(n=4)
Procedure longer than 15mins
(n=10)

Lost to follow-up (n= 7 ),
Withdraw inform content(n=3)
Procedure longer than 15mins
(n=11)

**Analysis**

Analysed  (n=111  )

Analysed  (n=97 )

Analysed  (n=96 )

**Fig 1. Consolidated standards of reporting trials (CONSORT) 2010 flow diagram of patients' distribution.**

**Table 1. Baseline characteristics.**

| Characteristics | RT (n = 111) | P (n = 97) | R+P (n = 96) | P Value |
|---|---|---|---|---|
| Age | 44±13 | 46±13 | 47±13 | 0.246 |
| Men (n, %) | 50 (45%) | 48 (49%) | 49 (51%) | 0.666 |
| ASA | | | | 0.306 |
| I | 101(91.0%) | 86(88.7%) | 91(94.8%) | |
| II | 10(9%) | 11(11.3%) | 5(5.2%) | |
| Height(cm) | 161.6±6.5 | 161.6±6.5 | 163.0±6.9 | 0.172 |
| Weight(kg) | 59.5±9.2 | 61.1±9.7 | 60.2±9.0 | 0.520 |
| BMI(kg/m$^2$) | 22.8±2.8 | 23.5±2.9 | 22.6±2.6 | 0.084 |
| MAP(mmHg) | 86±12 | 88±10 | 87±8 | 0.058 |
| HR(bpm) | 75±12 | 75±12 | 74±12 | 0.304 |
| RR(bpm) | 17±2 | 17±3 | 17±2 | 0.130 |
| SpO$_2$(%) | 98±1 | 99±1 | 99±1 | 0.054 |

Note: Data are presented as mean ± standard deviation (SD) or number (percentage).

Abbreviations: BMI, body mass index; ASA, American Society of Anesthesiologists; MAP, mean artery pressure; HR, heart rate; RR, respiratory rate; SpO2, pulse oxygen saturation.

31.3%), and the lowest incidence was in the RT group (n = 14, 12.6%), with statistical differences among the three groups in pairwise comparison (Table 2).

### Secondary outcomes

The incidence of hypotension requiring treatment was, in descending order, in the P group (n = 19, 19.6%), the R+P group (n = 6, 6.3%), and the RT group (n = 2, 1.8%), the P group was

**Table 2. Primary and secondary outcomes in the population.**

| | RT | P | R+P | P value |
|---|---|---|---|---|
| Primary outcome | | | | |
| MAP decrease 20%, MAP<65mmHg, No. (%) | 14(12.7%) | 55(56.7%) | 30(31.3%) | <0.001[a, b,c] |
| Secondary outcomes | | | | |
| Duration of MAP decrease 20%(minutes) | 0.76±1.92 | 4.43±4.13 | 1.21±2.55 | <0.001[a,b] |
| MAP decrease 30% | 2(1.8%) | 19(19.6%) | 6(6.3%) | <0.05[a,b] |
| Duration of MAP decrease 30%(minutes) | 0.07±0.37 | 1.34±2.73 | 0.71±0.69 | <0.001[a,b] |
| Time of LOC (minutes) | 10.07±3.94 | 9.40±3.66 | 8.59±2.38 | 0.012[c] |
| Steward score | | | | |
| 10minutes | 5.60±0.93 | 5.66±1.12 | 5.96±0.41 | 0.034[b,c] |
| 20minutes | 5.90±0.45 | 5.95±0.42 | 6.00±0 | 0.051 |
| Orientational force | | | | |
| 10minutes | 9.03±2.46 | 9.02±2.80 | 9.85±1.04 | 0.020[b,c] |
| 20minutes | 9.96±0.27 | 9.89±1.02 | 9.98±0.14 | 0.516 |
| Ataxia rating | 2.74±3.47 | 4.09±5.02 | 1.30±2.22 | <0.001[b,c] |
| Satisfaction rating | | | | |
| Operators | 3.53±0.84 | 3.87±0.44 | 3.95±0.22 | <0.001[a,b] |
| Anesthesiologists | 3.52±0.84 | 3.85±0.39 | 3.96±0.17 | 0.002[a,b] |
| patients | 3.94±0.22 | 3.86±0.45 | 3.97±0.17 | 0.086 |

"a" indicates RT group compared with P group, "b" indicates P group compared with R+P group, and "c" indicates RT group compared with R+P group.

MAP: Mean Arterial Pressure; LOC: loss of consciousness

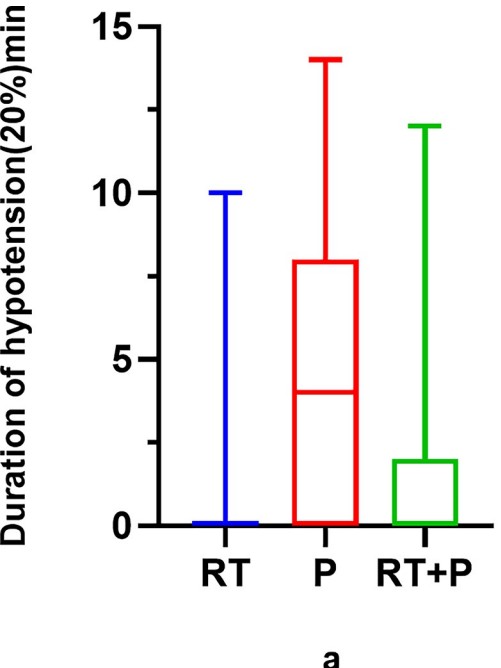
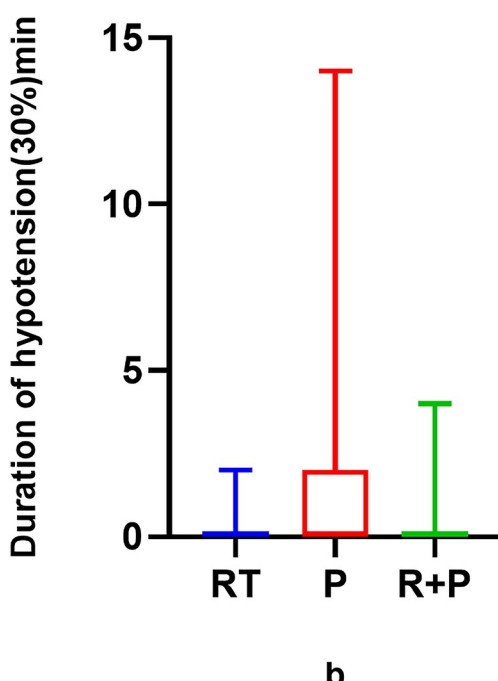

**Fig 2. Duration of blood pressure decrease.** RT represents the Remimazolam tosylate group, P represents the propofol group, RT+P represents the Remimazolam combined with propofol group.

statistically different from the R+P and RT groups, respectively (see Table 2). The duration of blood pressure decrease was the longest in the P group (Fig 2).

Among 304 patients, the success rate of sedation was not statistically different among the three groups, sedation failure occurred in only one patient, in the RT group, where the patient had a MOAA/S score of 4 after five consecutive additional doses of remimazolam and did not achieve the required depth of sedation to start the examination, which was eventually remedied by propofol to complete the procedure. Time to LOC was longest in the RT group (83 ±40s) and comparable in the P group (75±29s) and the R+P group (73±18s); The mean MOAA/S scores after drug administration in the three groups are shown in Fig 3.

The three groups had the shortest recovery time in the R+P group, with a statistically significant difference compared with the RT group (8.59±2.38minutes, vs10.07±3.94minutes). The steward score and orientation score at 10 minutes postoperatively were better in the R+P group than in the RT and P groups. Modified Bromage score and head-up assessment were better in the P and R+P groups than in the RT group at 10 minutes postoperatively, $P< 0.001$ (Fig 4A and 4B), while there was no statistical difference at 20 minutes postoperatively. Scores of the ataxia scale were lower in the R+P group than in the RT and P groups (1.30±2.22, vs. 2.74±3.47, vs. 4.09± 5.02), $P< 0.001$.

The results of the endoscopist satisfaction and anesthesiologist satisfaction were higher in the P and R+P groups than in the RT group, and there was no statistical difference in the comparison between the P and R+P groups, $P<0.05$; there was no statistical difference in patient satisfaction for all three groups.

## Safety results

Any adverse events (including the incidence of hypotension) occurred in 68 cases in the RT group, 91 cases in the propofol group, and 41 cases in the R+P group, with a significant

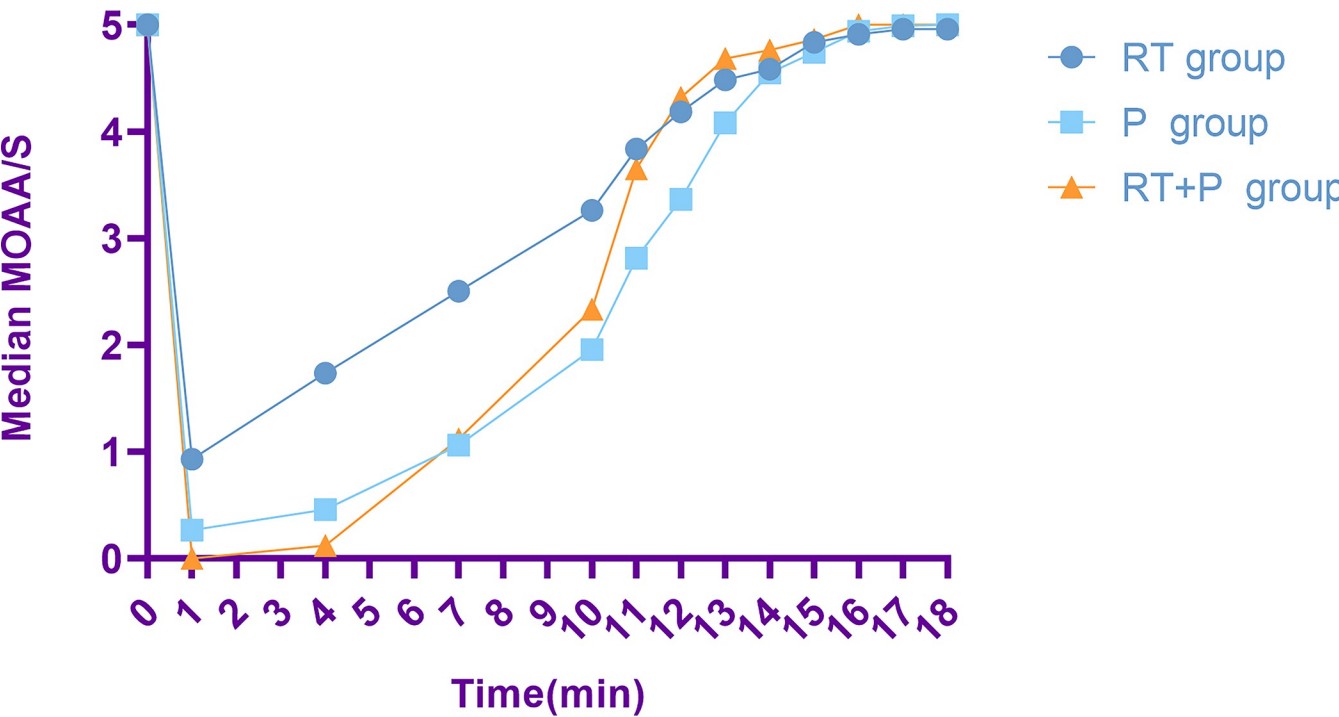

**Fig 3. Mean MOAA/S score.** RT represents the remimazolam tosylate group, P represents the propofol group, RT+P represents the remimazolam combined with propofol group.

difference in pairwise comparison, P<0.001. A total of 53 patients had injection pain in the P group, while 6 patients in the RT group, and 11 patients in the R+P group, with a significantly lower incidence of injection pain in the RT and R+P groups than in the P group, P<0.001, with no statistical difference between the RT and R+P groups. Respiratory adverse events (including hypoxemia, respiratory depression, and upper airway obstruction) were 17 cases in the RT group, 22 cases in the propofol group, and 7 cases in the R+P group, with significant differences between the RT and R+P groups, $P = 0.012$. The incidence of body movement was

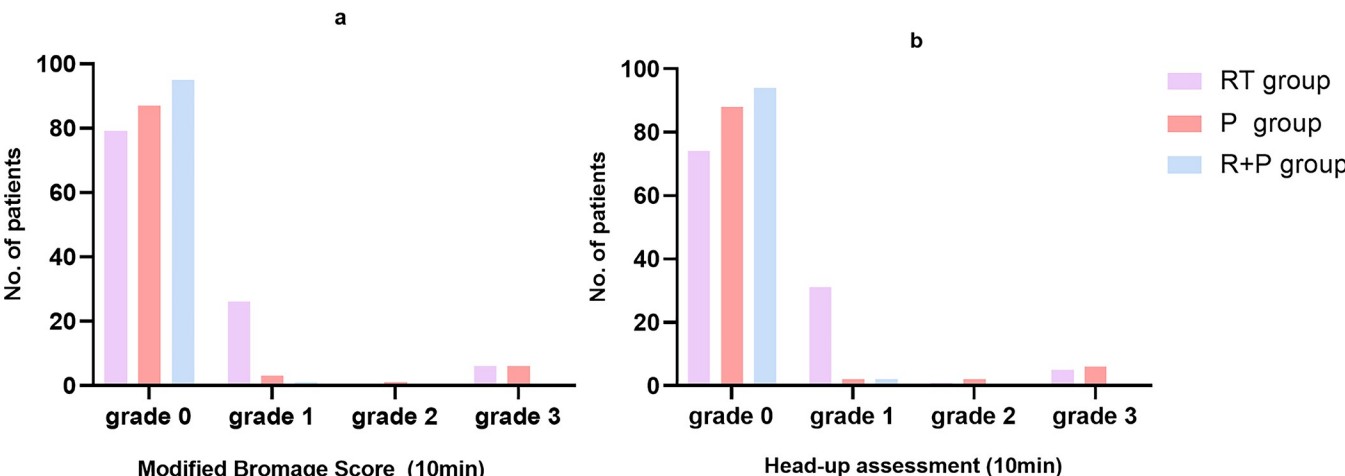

**Fig 4. Modified bromage score and head-up assessment.** RT represents the remimazolam tosylate group, P represents the propofol group, RT+P represents the remimazolam combined with propofol group. The lower the grade, the higher the quality of patient recovery.

**Table 3. Adverse event analysis.**

| Name, No.(%) | RT | P | R+P | P |
|---|---|---|---|---|
| AnyAE | 68(61.3) | 91(96.8) | 41(42.7) | <0.001[a,b,c] |
| Respiratory AE | 17(15.3) | 22(22.7) | 7(7.3) | 0.012[b] |
| SPO₂<95% | 4(3.6) | 9(9.3) | 4(4.2) | |
| RR<8bpm | 9(8.1) | 7(7.2) | 0 | |
| RR = 0 | 4(3.6) | 3(3.1) | 0 | |
| Airway obstruction | 0 | 3(3.1) | 3(3.1) | |
| Injection pain,No.(%) | 6(5.4) | 53(54.6) | 11(11.5) | <0.001[a,b] |
| Body movement | 29(26.1) | 10(10.3) | 12(12.5) | 0.004[a,c] |
| Chocking and coughing | 4(3.6) | 1(1.0) | 0 | 0.107 |
| Nausea and vomiting | 2(1.8) | 0 | 0 | 0.174 |
| hiccup | 2(1.8) | 0 | 2(2.1) | 0.381 |
| Bradycardia | 1(0.9) | 3(3.1) | 1(1.0) | 0.396 |

"a" indicates RT group compared with P group, "b" indicates P group compared with R+P group, and "c" indicates

AE: Adverse Event

significantly higher in the RT group (29 cases) than in the P group (10 cases) and the R+P group (12 cases), $P = 0.004$, whereas no difference between the P group and the R+P group, $P = 0.685$. The remaining adverse events were not statistically different among the three groups, as shown in Table 3.

## Discussion

Recently, endoscopy sedation has been extensively used, and progressively more clinicians prefer propofol sedation due to the slow onset and longer action of benzodiazepines [3, 5]. Nonetheless, the management of propofol is considerably strict by the exceedingly narrow therapeutic window of propofol, only anesthesiologists or professionally trained medical personnel can administer propofol sedation in many countries, which will undoubtedly increase the medical labor cost and the financial burden on patients [3, 21]. In our study, all patients were successfully sedated in the R+P group, with no statistical difference in the incidence of body movement compared with the P group (10.3% *vs*. 12.5%), suggesting that the combination of remimazolam tosylate and propofol provides comparable efficacy to propofol monotherapy, avoids the possible inadequate depth of sedation with remimazolam monotherapy, improves endoscopist satisfaction, and reduces the incidence of adverse events and improve the safety of patients. Because of the small dose of propofol administered (0.5 mg/kg), patient safety can be similarly assured even when used by non-anesthesiologists, which may have the opportunity to address the current dilemma of sedation for gastrointestinal endoscopy.

In a multicenter study of upper gastrointestinal endoscopy [22], the incidence of hypotension in the remimazolam tosylate group was significantly lower than that in the propofol group (13.04% vs 42.86%), which is generally consistent with our findings, and the incidence of hypotension in the R+P group in our study was 31.3%, which was lower than that in the propofol group, The possible reason for this is that propofol decrease blood pressure by affecting endothelial cell function, vascular calcium signaling, and sympathetic nervous system activity [23–25], as well the effect of remimazolam on cardiac output (CO) is smaller [9]. Furthermore, it is worth mentioning that the duration of hypotension in the R+P group (1.21±2.25minutes) was within 5 minutes, co-administration may be able to reduce the likelihood of adverse outcomes in patients and has greater value for clinical application.

Disparate with some clinical trials, we did not conclude that respiratory adverse events in the RT group were statistically different from those in the P group [16, 26, 27]. We regard that the possible reasons are: 1) We selected patients with a mean BMI of 22~24, which is in the normal range and the chance of adverse respiratory events is lower, accordingly reducing the difference among the three groups; 2) The sample size was calculated by the incidence of hypotension, the true results on respiratory adverse events could not be obtained with the sample size of this study, expanding the sample size may get accurate results.

The incidence of injection pain in propofol sedation is a high and large individual difference, with a mean pain score of up to $4.82 \pm 1.73$, and individual patients can even reach 10, which is particularly impressive to patients even called "the most painful experience" [28–30]. Propofol injection pain may be related to the concentration of free propofol [31], and remimazolam is water-soluble, as a consequence that the incidence of injection pain is remarkably reduced [32, 33]. Our study found that the incidence of injection pain was significantly lower in the R+P group than in the P group, which is similar to the findings of another prospective study in an abortion population, which concluded that early administration of 0.1 mg/kg of remimazolam prevented injection pain from propofol with similar effects to the addition of lidocaine [34]. Thus, the combination of the two drugs can optimize the administration process and reduce unpleasant memories for patients.

The mean time to discharge for sedation of propofol and remimazolam was 24 minutes and 21 minutes, respectively [35].While outpatients usually need to be monitored for at least 30 minutes before they can be discharged [36], physicians involved in sedation usually tend to choose the drug that has the least impact on recovery to avoid danger occurring when patients are discharged, thus the quality of recovery is important in outpatient procedures. However, some studies have shown that repeated exposure to general anesthesia may affect the patient's fine-motor function [37], We consider that sedative drugs might affect the fine-motor function in the short term and increase the risk of patients when leaving the hospital, so we also assessed the recovery of the fine-motor function based on general recovery, but because there is no uniform scale for assessing fine-motor assessment, part of the International Cooperative Rating Scale (ICARS) was selected to assess. The higher the score, the more severe the ataxia was. Our results showed that the total ataxia severity was lower in the RT group (2.74±3.47) and the R+P group (1.30±2.22) than in the P group (4.09±5.02), indicating that the effect of remimazolam tosylate and propofol in combination on fine-motor function was less than that of propofol.

It should be noted that the dose selection of the two drugs in the R+P group was made by referring to the relevant literature [38]. In this study, four dose groups were set up by pre-experimentation, 0.1 mg/kg of remimazolam + 0.5 mg/kg of propofol, 0.1 mg/kg of remimazolam + 1 mg/kg of propofol, 0.15 mg/kg of remimazolam + 0.5 mg/kg of propofol, and 0.15 mg/kg of remimazolam + 1 mg/kg of propofol, respectively. According to the preliminary statistical analysis, the combination of 0.1mg/kg of remimazolam + 0.5mg/kg of propofol can meet the adequate depth of sedation, while other doses can meet the sedation requirement, but the incidence of hypotension also increases. As a consequence, we set the dose of the drug combination for the R+P group to 0.1mg/kg of remimazolam + 0.5mg/kg of propofol. The additional drug chosen for the R+P group was propofol because a rapid decrease in blood pressure generally occurs 2 minutes after induction, after which blood pressure may continue to fall but with a relatively flat trend [39], we believe that the administration of propofol during maintenance does not tend to increase the incidence of hypotension and ensures better sedation, but it does not mean that this is the only option, and further studies are needed to investigate the different combination doses and administration methods of the two drugs.

### Limitation

In this study, the combination of propofol and remimazolam tosylate was used, a dose escalation test was conducted in a pre-experimental by the sequential method in the selection of the dose of the two combinations, and the MOAA/S$\leq$1 for remimazolam 0.1 mg/kg+propofol 0.5 mg/kg could be satisfied., the dose was not tested with a large sample size and may not be the optimal dose for the combination of the two drugs. Therefore, the optimal dose needs to be further investigated.

## Conclusion

Co-administration of remimazolam tosylate and propofol could reduce adverse events in upper gastrointestinal endoscopy, the success rate of sedation was the same with propofol, and had higher endoscopist satisfaction than remimazolam monotherapy, suggesting that the combination of the two drugs is feasible and may have greater clinical value in outpatient examinations.

## Supporting information

**S1 Checklist. This is the CONSORT checklist.**
(DOC)

**S1 Protocol. This is the Chinese version of the research proposal.**
(DOCX)

**S2 Protocol. This is the English version of the research proposal.**
(PDF)

**S1 Data.**
(XLSX)

## Acknowledgments

We acknowledge the nurses who dispensed the drugs and to the physicians collected the data for the trial at each center.

## Author Contributions

**Conceptualization:** Mengchang Yang.

**Data curation:** Ai Wei, Shijin Ma, Xiaojun Wang, Jianxiong Wu, Shuzhi Zhou, Yanfang Deng, Xinquan Liu.

**Formal analysis:** Yuzhe Dou, Dongming Li.

**Funding acquisition:** Shuzhi Zhou.

**Investigation:** Shijin Ma.

**Project administration:** Dongming Li.

**Resources:** Yanfang Deng.

**Software:** Yuzhe Dou.

**Supervision:** Mengchang Yang.

**Validation:** Xinquan Liu.

**Writing – original draft:** Ai Wei.

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
