## [Decision Letter · Decision Letter 0]

30 May 2023

PONE-D-23-05242The safety and efficacy of remimazolam tosylate combined with propofol in upper gastrointestinal endoscopy: a multicenter, randomized clinical trialPLOS ONE

Dear Dr. Yang,

Thank you for submitting your manuscript to PLOS ONE. After careful consideration, we feel that it has merit but does not fully meet PLOS ONE’s publication criteria as it currently stands. Therefore, we invite you to submit a revised version of the manuscript that addresses the points raised during the review process.

We look forward to receiving your revised manuscript.

Kind regards,

Ioannis Savvas, DVM, Ph.D.

Academic Editor

PLOS ONE

Journal Requirements:

2. We note that the status of the clinical trial entry (NCT05429086) is 'not yet recruiting'. Please ensure to update the entry to the current status of the clinical trial.

3. We note that in the current version of your manuscript Figure 1 is "Fig 1. Duration of blood pressure decrease". Please note that you must upload the completed CONSORT flowchart as figure 1 of your manuscript.

4. The information in the registry entry suggests that your trial was registered after patient

recruitment began. PLOS ONE strongly encourages authors to register all trials before recruiting the first participant in a study.

a) your reasons for your delay in registering this study (after enrolment of participants started);

b) confirmation that all related trials are registered by stating: “The authors confirm that all ongoing and related trials for this drug/intervention are registered”.

Please also ensure you report the date at which the ethics committee approved the study as well as the complete date range for patient recruitment and follow-up in the Methods section of your manuscript.

Reviewers' comments:

Reviewer's Responses to Questions

**Comments to the Author**

1. Is the manuscript technically sound, and do the data support the conclusions?

Reviewer #1: Partly

Reviewer #2: Yes

2. Has the statistical analysis been performed appropriately and rigorously? 

Reviewer #1: Yes

Reviewer #2: Yes

3. Have the authors made all data underlying the findings in their manuscript fully available?

Reviewer #1: Yes

Reviewer #2: Yes

4. Is the manuscript presented in an intelligible fashion and written in standard English?

Reviewer #1: Yes

Reviewer #2: Yes

5. Review Comments to the Author

Reviewer #1: A three-arm randomized clinical trial was conducted which aimed to investigate the safety and efficacy of remimazolam tosylate with propofol as a sedative for upper gastrointestinal endoscopy. The incidence of hypertension significantly differed in the three arms. The proportion with body movement was significantly higher in the RT arm compared to the other two arms.

Minor revisions:

1- Abstract: Define all abbreviations at first occurrence.

2- Abstract: Provide pairwise comparisons to show which arms differed with respect to hypotension rates.

3- Randomization and Blinding: If block randomization was used, indicate the block size.

4- Sample size calculation and statistical analysis (p. 8): State the statistical testing method which achieves 80% power.

5- Sample size calculation and statistical analysis: (p. 9): Specify the nonparametric tests used for quantitative data.

6- Table 1: Explain the type of summary data presented for ASA.

7- State the statistical testing method used for pairwise comparisons when the overall ANOVA was significant. Indicate if chi-square tests used for pairwise comparisons were corrected for multiple comparisons.

8- Indicate if adverse events were collected according to a standardized method.

9- Table 3: Provide the percentages that correspond to the counts.

10- Fig 1: Label the units of time on the y-axis.

11- To assist in the review process, add line numbering to the document.

Reviewer #2: The authors reported the safety and efficacy of reminazolam tosylate combined with propofol in upper gastrointestinal endoscopy in a multiple, randomized clinical trial. It is interesting and provides useful information for the clinical application of reminazolam tosylate and propofol in upper gastrointestinal endoscopy. The main concerns are listed in the following.

1. Previous study has reported that Remimazolam Tosylate Combined with Low-Dose Propofol Improves Sedation and Safety in Hysteroscopy (Drug Des Devel Ther. 2022 Nov 29;16:4101-4108.) . What is the difference between the study and your srtudy?

2. The adverse events should be listed in the methods of ABSTRACT.

3. MOAA/S score was used to evaluate the anesthesia depth. However, it is difficult to indicate that the anesthesia depth is consistent among groups. Maybe, Bis is better.

4. In this study, there are three groups. It means that you have to doχ2 tests twice or three times, so the p values cannot be set at 0.05. It should be 0.025 or 0.017.

6. PLOS authors have the option to publish the peer review history of their article (what does this mean?). If published, this will include your full peer review and any attached files.

Reviewer #1: No

Reviewer #2: No

---

## [Author Response · Author response to Decision Letter 0]

27 Jun 2023

To academic editor:

Reply: Thank you for your reminder. We have revised the manuscript according to the magazine format requirements

2. We note that the status of the clinical trial entry (NCT05429086) is 'not yet recruiting'. Please ensure to update the entry to the current status of the clinical trial.

Reply：We have updated the research information at the Clinical Trials.

3. We note that in the current version of your manuscript Figure 1 is "Fig 1. Duration of blood pressure decrease". Please note that you must upload the completed CONSORT flowchart as figure 1 of your manuscript.

Reply：Information of figure has been modified

4. The information in the registry entry suggests that your trial was registered after patient

recruitment began. PLOS ONE strongly encourages authors to register all trials before recruiting the first participant in a study.

a) your reasons for your delay in registering this study (after enrolment of participants started);

b) confirmation that all related trials are registered by stating: “The authors confirm that all ongoing and related trials for this drug/intervention are registered”.

Please also ensure you report the date at which the ethics committee approved the study as well as the complete date range for patient recruitment and follow-up in the Methods section of your manuscript.

Reply: Thank you for your reminder. We have noticed that the recruitment time for patients in the article was before the trial registration. We apologize for the misunderstanding caused by our pen mistake. The actual recruitment time for patients was July 1, 2022, and this information has been modified in the article; We have confirmed that all relevant trials have been registered (Line 98) and reported the time when the ethics committee approved the study, as well as the complete time for recruiting patients and completing follow-up. (Line 101-102)

To Reviewer #1：

1- Abstract: Define all abbreviations at first occurrence

Reply: Already defined. (Line 30-31)

2- Abstract: Provide pairwise comparisons to show which arms differed with respect to hypotension rates.

Reply：Three arms of pairwise comparisons have statistical differences, which have been noted in the article(Line 37-38)

3- Randomization and Blinding: If block randomization was used, indicate the block size.

Reply：Already modified，block of size of 6. (Line 120)

4- Sample size calculation and statistical analysis (p. 8): State the statistical testing method which achieves 80% power.

Reply：A sample size of 229 achieves 80% power to detect an effect size of 0.2051 using a 2 degrees of freedom Chi-Square Test with a significance level (alpha) of 0.05000.(Line 205-207)

5- Sample size calculation and statistical analysis: (p. 9): Specify the nonparametric tests used for quantitative data.

Reply: We have annotated the specific methods of nonparametric testing in the article (Kruskal-Wallis test) (Line 215)

6- Table 1: Explain the type of summary data presented for ASA.

Reply: We have made modifications in the article.(Table 1)

7- State the statistical testing method used for pairwise comparisons when the overall ANOVA was significant. Indicate if chi-square tests used for pairwise comparisons were corrected for multiple comparisons. 

Reply: if the difference was significant, the Student-Newman-Keuls q test was further used for pairwise comparison of each group.(Line 216-218). In �2 test , multiple comparisons of the enumeration data (P group to R group, P group to R+P group, R group to R+P group) were performed, and the α lever was set at 0.017, following Bonferroni adjustment. (Line 219-222)

8- Indicate if adverse events were collected according to a standardized method.

Reply: We have a unified definition for all adverse events and collect them strictly according to the definition, and data is collected uniformly by the evaluation researchers in each center. We have clearly marked it in the manuscript. (Line 179-191)

9- Table 3: Provide the percentages that correspond to the counts.

Reply: Table 3 has added corresponding percentages.

10- Fig 1: Label the units of time on the y-axis.

Reply: Already modified

11- To assist in the review process, add line numbering to the document.

Reply：The manuscript has been added with a line number.

To Reviewer #2:

1- Previous study has reported that Remimazolam Tosylate Combined with Low-Dose Propofol Improves Sedation and Safety in Hysteroscopy (Drug Des Devel Ther. 2022 Nov 29;16:4101-4108.) . What is the difference between the study and your study? 

Reply：First of all, our research is different from the purpose of this study. The purpose of this study is to explore the incidence of hypoxemia among the three groups, while our research purpose is to explore the incidence of hypotension among the three groups. Previous studies have shown that hypotension is the most common adverse event when propofol is used for endoscopic sedation, and the degree and duration of Hypotension may be related to organ damage or even mortality in patients（Hypotension during propofol sedation for colonoscopy: a retrospective exploratory analysis and meta-analysis；British Journal of Anaesthesia, 128 (4): 610e622 (2022). Therefore, hypotension should be considered as a serious adverse event in digestive endoscopy, which deserves more clinical attention; Secondly, our research design is more comprehensive. This study is a single center study with a small sample size, while our study is a multicenter study with a larger population and more observed outcome indicators. Our research results include all the results of this study, and we have conducted a deeper exploration of patient recovery and increased satisfaction surveys, which can provide guidance for clinical applications; Finally, our research focuses on the population preparing for gastrointestinal endoscopy, and to our knowledge, there are currently no studies on the combination of remimazolam and propofol in gastrointestinal endoscopy.

2- The adverse events should be listed in the methods of ABSTRACT. 

Reply：Adverse events have been added in the summary section. (Line 33-35)

3- MOAA/S score was used to evaluate the anesthesia depth. However, it is difficult to indicate that the anesthesia depth is consistent among groups. Maybe, Bis is better. 

Reply: Thank you very much for your suggestion. In response to this issue, the MOAA/S score is currently a commonly used standard for determining the depth of sedation in outpatient settings. It is simple, fast, and can guide clinical medication administration; Secondly, the intravenous Narcotic with the best correlation with BIS is propofol, and benzodiazepines have a worse correlation with BIS than propofol. In our previous research, we explored the correlation between Remimazolam and BIS, but unfortunately, we found that when more patients were given larger doses of Remimazolam, even though the depth of sedation was enough, the BIS value was still high, So we did not consider using BIS for guidance on anesthesia depth in this study.

4- In this study, there are three groups. It means that you have to do χ2 tests twice or three times, so the p values cannot be set at 0.05. It should be 0.025 or 0.017.

Reply：Thank you for your suggestion. The P-value has been corrected. (Line 219-222)

---

## [Decision Letter · Decision Letter 1]

19 Jul 2023

The safety and efficacy of remimazolam tosylate combined with propofol in upper gastrointestinal endoscopy: a multicenter, randomized clinical trial

PONE-D-23-05242R1

Dear Dr. Yang,

We’re pleased to inform you that your manuscript has been judged scientifically suitable for publication and will be formally accepted for publication once it meets all outstanding technical requirements.

Kind regards,

Ioannis Savvas, DVM, Ph.D.

Academic Editor

PLOS ONE

Additional Editor Comments (optional):

Reviewers' comments:

Reviewer's Responses to Questions

**Comments to the Author**

1. If the authors have adequately addressed your comments raised in a previous round of review and you feel that this manuscript is now acceptable for publication, you may indicate that here to bypass the “Comments to the Author” section, enter your conflict of interest statement in the “Confidential to Editor” section, and submit your "Accept" recommendation.

Reviewer #1: (No Response)

Reviewer #2: All comments have been addressed

2. Is the manuscript technically sound, and do the data support the conclusions?

Reviewer #1: Yes

Reviewer #2: Yes

3. Has the statistical analysis been performed appropriately and rigorously? 

Reviewer #1: Yes

Reviewer #2: Yes

4. Have the authors made all data underlying the findings in their manuscript fully available?

Reviewer #1: Yes

Reviewer #2: Yes

5. Is the manuscript presented in an intelligible fashion and written in standard English?

Reviewer #1: Yes

Reviewer #2: (No Response)

6. Review Comments to the Author

Reviewer #1: Minor revisions:

All comments have been addressed. However, there are two typographical errors.

1- Line 31: Typographical error: R+P group

2- Line 223: Typographical error: alpha level

Note: Line numbers refer to those in the track changes of version 1.

Reviewer #2: The authors have answered all my questions. I have no more question. I think it is acceptable now.

7. PLOS authors have the option to publish the peer review history of their article (what does this mean?). If published, this will include your full peer review and any attached files.

Reviewer #1: No

Reviewer #2: No

---

## [Editor Report · Acceptance letter]

26 Jul 2023

PONE-D-23-05242R1 

The safety and efficacy of remimazolam tosylate combined with propofol in upper gastrointestinal endoscopy: a multicenter, randomized clinical trial 

Dear Dr. Yang:

I'm pleased to inform you that your manuscript has been deemed suitable for publication in PLOS ONE. Congratulations! Your manuscript is now with our production department. 

Kind regards, 

on behalf of

Prof. Ioannis Savvas 

Academic Editor

PLOS ONE